# AliMarko: A Pipeline for Virus Identification Using an Expert-Guided Approach

**DOI:** 10.3390/v17030355

**Published:** 2025-02-28

**Authors:** Nikolay Popov, Ignat Sonets, Anastasia Evdokimova, Maria Molchanova, Vera Panova, Elena Korneenko, Alexander Manolov, Elena Ilina

**Affiliations:** 1Research Institute for Systems Biology and Medicine (RISBM) of Rospotrebnadzor, 117246 Moscow, Russia; 2Martsinovsky Institute of Medical Parasitology, Tropical and Vector Borne Diseases, Sechenov First Moscow State Medical University, 119435 Moscow, Russia; 3Independent Researcher, 393410 Tambov, Russia; 4Moscow Center for Advanced Studies, Kulakova Str. 20, 123592 Moscow, Russia; 5Department of Bioengineering and Bioinformatics, Lomonosov Moscow State University, Leninskie Gory 1, 119234 Moscow, Russia

**Keywords:** virus discovery, phylogenetic analysis, virome, automated pipeline

## Abstract

Viruses are ubiquitous across all kingdoms of cellular life, posing a significant threat to human health, and analyzing viral communities is challenging due to their genetic diversity and lack of a single, universally conserved marker gene. To address this challenge, we developed the AliMarko pipeline, a tool designed to streamline virus identification in metagenomic data. Our pipeline uses a dual approach, combining mapping reads with reference genomes and a de novo assembly-based approach involving an HMM-based homology search and phylogenetic analysis, to enable comprehensive detection of viral sequences, including low-coverage and divergent sequences. We applied our pipeline to total RNA sequencing of bat feces and identified a range of viruses, quickly validating viral sequences and assessing their phylogenetic relationships. We hope that the AliMarko pipeline will be a useful resource for the scientific community, facilitating the interpretation of viral communities and advancing our understanding of viral diversity and its impact on human health.

## 1. Introduction

Viruses are inherently complex and challenging objects of study due to their genetic heterogeneity and the absence of a single, universal single-copy genes that could serve as a marker for their identification and classification (similar to the 16S ribosomal RNA gene in bacteria) [1]. Instead, they feature various hallmark genes that permit the identification of particular known viral families [2,3]. The amplification of hallmark genes poses a significant challenge, primarily due to their high sequence divergence and large number of hallmark genes. These limitations promote the use of metagenomic approaches for analyzing viral communities, which enable researchers to overcome the constraints of gene amplification [4].

Metagenomic analysis of viral communities is further complicated by the low abundance of viral sequences, making them more susceptible to contamination and increasing the risk of false-positive results. Contamination in viral metagenomics, such as sequences from reagents and library preparation kits, can distort research outcomes and lead to false associations with diseases [5]. For example, contaminating Acanthocystis turfacea chlorella virus 1 [6] and Xenotropic murine leukemia virus [7] sequences in human samples caused the misassociation of them with alterations in cognitive function and chronic fatigue syndrome, respectively [8,9]. For instance, Asplund et al., 2019, revealed an association between a wide range of viral sequences and reagents used in library preparation [10]. Other work reports contaminations coming from RNA purification kits [11].

Several widely used utilities are available for detecting viral sequences in metagenomes [12]. A commonly employed approach involves local sequence alignment against a reference database using the BLAST [13] tool to identify viral sequences or utilizing BLAST-based tools such as VirusSeeker [14]. Another widely used method is the mapping of short sequences to viral genomes, with BWA [15] and Bowtie2 [16] being two popular tools. BWA/Bowtie2 are good choices for detecting viral sequences, offering high sensitivity and fast processing times, but they are reference-dependent and may not detect novel viruses. Additionally, Hidden Markov Models (HMMs) are used to recognize viral protein sequences, with HMMer3 being the most widely used instrument for this approach [17]. HMMer3 is a powerful tool that can detect novel or divergent viruses, but it requires careful model selection to avoid false positives. A similar method involves the use of Position-Specific Scoring Matrices (PSSMs), which enable efficient computational searches for ungapped matches [18]. An alternative approach to detect viral sequences involves analyzing the k-mer (short sub-sequences of fixed length) spectrum. Kraken2 is a prominent tool that implements this approach [19], offering fast and efficient processing times and reference-independent detection, but it may be less sensitive than BWA/Bowtie2 or HMMer3.

Some tools use a combination of methods to leverage the advantages of different approaches. VirSorter2 uses a combination of HMM and common features analysis (like GC content and gene density) [20]. VirFinder uses a combination of k-mers and SVM and returns the likelihood of being a viral sequence [21]. VirID is a comprehensive analysis tool that performs RNA virus detection and characterization through contig assembly, homology search, and phylogenetic analysis, returning taxonomic information on identified viral contigs [22].

In recent years, several tools have been developed for identifying virus sequences using recurrent or convolutional neural networks, such as ViBE [23] and DeepVirFInder [24]. These models can be computationally intensive, which may limit their applicability.

To summarize, a number of tools have already been developed for the automatic identification of viral sequences. However, the identification of virus sequences is a complex process, and the ambiguity in setting appropriate thresholds can pose significant challenges for practicing biologists and epidemiologists. Researchers often find themselves needing to verify the findings of automated tools to ensure the reliability of the results. Our approach focuses on providing a solution that helps researchers confidently assess and validate the outputs of automated systems and filter false-positive findings.

In the field of virus metagenomics, bats are recognized as an important reservoir for a broad spectrum of viruses, encompassing *Lyssaviridae*, *Filoviridae*, *Paramyxoviridae*, and other families [25]. Their characteristics, such as long lifespan, social behavior, and wide geographic distribution, make them a natural hub for this diverse range of viruses. Metagenomic sequencing has enabled researchers to explore the viral diversity of bats, revealing a vast number of unknown viruses [26]. The study of bat viruses is crucial for early detection and monitoring of potential health threats.

## 2. Materials and Methods

The AliMarko pipeline consists of preprocessing, reference-based mapping, de novo assembly, and phylogenetic analysis steps to identify viral sequences in metagenomic data.

### 2.1. Read Filtration

Kraken2 is used to deplete reads of cellular organisms in order to lower false-positive viral identifications. We built a custom database that includes the “UniVec_Core”, “archaea”, “bacteria”, “fungi”, “human”, “plant”, and “protozoa” libraries, and it is available as part of AliMarko. Kraken2 is launched with a confidence parameter set to default 0.7 and the parameter “--unclassified-out” to write unclassified reads to fastq files for further analysis.

Reads deduplication is performed with fastp [27]. Reads with quality scores less than 15 are excluded using samtools view.

### 2.2. Read Mapping and Analysis

Read mapping is performed with BWA-MEM [28]. Mappings with a quality score of less than 20 and match percentage with a heuristic threshold of 63% of matching are eliminated. The coverage width and mean mapping quality of each reference genome are calculated using samtools.

Mapping visualization is performed with BamSnap [29]. We modified the source code of BamSnap to allow it to visualize full sequences. Modified version is available at: https://github.com/NJJeus/bamsnap (accessed on 2 June 2023).

### 2.3. HMM Analysis

De novo assembly is performed using SPAdes [30] with the -meta option or MEGAHIT [31]. HMM analysis is conducted using the pyHMMer library [32] and Viral Minion DB HMM [33,34] on six-frame translated nucleotide sequences.

For each HMM, a threshold score was determined as follows. All non-viral proteins (negative set) were selected from the Swiss-Prot database [35]. The hmmscan tool was then run on this set, and the highest score obtained was chosen as the threshold for further use. When performing analysis, only HMM matches with a score greater than the threshold are further considered. Regions of the contig that match the HMM profile are visualized using custom Python code.

Each contig matched by the HMM model with a score above the threshold undergoes a homology search using blastn against the utilized reference sequence database or another chosen one. The best result is output in an HTML report.

### 2.4. Phylogenetic Analysis

The phylogenetic analysis is conducted as follows: a region identified using HMM is extracted from the contig. Additionally, sequences from reference genomes are analyzed using the same HMM, and the identified regions are extracted. All these extracted regions are then added to a FASTA file. Multiple sequence alignment is performed using MAFFT [36]. A phylogenetic tree is constructed with FastTree [37], which infers approximately maximum-likelihood phylogenetic trees. FastTree provides local support values based on the Shimodaira-Hasegawa test to estimate the reliability of each split in the tree, which is depicted in a tree visualization. Trees are rooted using middle-point rooting via the BioPython Phylo library.

### 2.5. Task Management

The pipeline is written in Snakemake, a modern workflow management system that allows for simultaneous analysis of multiple samples, manages the use of multiple threads, and facilitates the installation of all required libraries.

### 2.6. Materials

We tested our pipeline on total NGS RNA sequencing of fecal samples from 26 bats captured in the Zvenigorodsky District of the Moscow Region [38]. RNA was extracted from the bat fecal samples using the QIAamp Viral RNA Mini Kit (Qiagen, Hilden, Germany). The extracted RNA was used for reverse transcription with Reverta-L reagents (AmpliSens, Moscow, Russia), and second-strand cDNA was synthesized using the NEBNext Ultra II Non-Directional RNA Second Strand Module (New England Biolabs, Ipswich, MA, USA, E6111L). Sequencing was performed on the Illumina MiSeq system to generate 250 bp paired-end reads. The data are available in SRA database by accessions SRR15540905, SRR15540904, SRR15534018, SRR15533116, SRR15533060, SRR15526222, SRR15525307, SRR15524477, SRR15524163, SRR15524530, SRR15508152, SRR15508267, SRR15508011.

To demonstrate the functionality of our computational pipeline, we created a small, simulated dataset. This dataset contains short reads emulating Illumina HS25 sequencing, generated using the art_illumina simulator (v2.5.8) with parameters -ss HS25 -p -l 150 [39]. The reference genomes used for simulation included a selection of bacterial species, the human genome, and the genomes of two viruses: Xanthomonas phage phiXv2 and Miniopterus bat Coronavirus 1 (Appendix A). Additionally, we incorporated sequences identified as belonging to the Murine leukemia virus, which were derived from bat sequencing data as an example of contamination.

To evaluate the accuracy and specificity of AliMarko, we created simulated metagenome samples. Using human nasopharyngeal swab sample SRR12183113 as a base, we estimated organism abundances with Kraken2, removed viral sequences, and re-generated a virus-free artificial sample using InSilicoSeq v2.0.1 [40] with parameters—model novaseq and—store_mutations. At the phylum level, the most abundant bacterial phyla were *Bacteroidota*, *Bacillota*, *Actinomycetota*, and *Pseudomonadota*.

For viral simulations, we used a reference containing 8 species from the families *Coronaviridae*, *Parvoviridae*, *Flaviviridae*, *Chaseviridae*, *Zobellviridae*, *Picobirnaviridae*, and *Phenuiviridae* (Appendix A. Sheet 2). Using the art_illumina simulator with the parameters mentioned above, we conducted five simulations with mutation rates for SNPs (0.01 to 0.17) and insertions (0.001 to 0.017), yielding mutated sequences with 79.05% to 98.72% identity to the original reference. For each mutation level, paired-end reads were simulated at 10× coverage and five datasets with varying genome coverage (from 250 to 2250 reads) were generated through subsampling. To standardize comparisons, reference genomes with >10% coverage width (AliMarko’s threshold) were used for read counting, and contigs with HMM hits above model-specific thresholds were analyzed.

For comparison with AliMarko, we utilized Kraken2 version 2.1.3 with the core_nt database (29 September 2024) and the plusPF database (14 May 2022).

AliMarko is available on GitHub https://github.com/NJJeus/AliMarko, accessed on 7 February 2025.

## 3. Results

### 3.1. Pipeline Overview

AliMarko processes FASTQ files by performing quality filtering and removing cellular reads. Following these preprocessing steps, two main types of analyses are conducted: (1) mapping to reference genomes and (2) de novo assembly of contigs followed by HMM analysis and phylogenetic analysis. The pipeline generates a comprehensive HTML report, featuring statistics and visualizations of the results for each module (Figure 1).

As a preliminary step, depletion of reads from cellular organisms is achieved using Kraken2 with a custom database (see Materials and Methods). This step is performed to reduce false positives, which can arise from nonspecific read mappings.

The next step is mapping to a database of reference genomes, which involves corresponding mapping results with a taxonomy metadata table and generating visualizations of the mapping. For this purpose, we use the reference genomes included in the Virus Metadata Resource (VMR) from the International Committee on Taxonomy of Viruses (ICTV) [41] as our database. VMR provides a universal taxonomic scheme of viruses, as well as access to current taxonomy, information about host organisms, and representative genomes, which we utilize as our metadata table.

The visualization of read mapping to reference genomes helps assess coverage width, uniformity of coverage depth, and single nucleotide substitutions density. This is especially useful if the virus in the sample is similar to those present in the genomic database.

We also provide information regarding the potential contaminant nature of a finding. For this, we use the information provided by (Asplund et al., 2019) [10]. If a genome in the mapping results exhibits high similarity to a laboratory-component-associated sequence, it is flagged in red and accompanied by a tooltip providing information about potential contaminant nature of the sequence.

In parallel to the mapping module, the HMM module is applied, which includes contig assembly and protein homology searching using HMM profiles. Each HMM profile has a predefined threshold to reduce false-positive results (see Materials and Methods). The HMM module aids experts in result interpretation by visualizing HMM matches and displaying a phylogenetic tree of the amino acid sequences matched by the HMM, including the matched contig sequences and corresponding reference sequences.

The pipeline conducts batch analysis and generates a comprehensive HTML report, allowing experts to explore specific sample reports for detailed insights.

### 3.2. AliMarko Reports

#### 3.2.1. One Sample Report

The HTML report provides tabular and graphical representations of the analysis results (Figure 2, S2 HTML).

At the top of the report, a table is displayed, which summarizes the results of the mapping module (Figure 2A). This table provides an overview of the mapping statistics with each row corresponding to a specific viral species. The table includes information such as the width and mean depth of coverage on the viral genome, host details, and the taxonomic classification of the virus. The viruses are arranged in descending order of coverage width.

Next, a second table is presented, which summarizes the results of the HMM module (Figure 2A). Each row in this table corresponds to a single HMM hit. The table includes details about each HMM hit, with each row containing the HMM name, associated meta-information, the contig name where the hit was found, and the hit score, including the raw score and the score normalized by the threshold score.

Following the tabular results, the report presents visualizations of the mapping results (Figure 2B). These visualizations are organized by the associated host. Each virus has a separate expandable frame that, when opened, displays a table with detailed mapping parameters for each segment of the virus. The table includes information such as the width and depth of coverage, as well as the number of identified SNPs. Below the table, a visualization of the mapping data is presented, offering a graphical representation of the read distribution across the genome and the localization of SNPs. For multi-segmented virus genomes, results are displayed for each segment in a row.

Below the mapping module details, the report presents the results of the HMM module in a tabular format (Figure 2B). Each contig is represented by a separate expandable frame, which, when opened, reveals a table summarizing the HMM hits against the contig. The table provides details such as the obtained score, meta-information on the HMM, and the normalized score. A visualization of HMM hits is provided for each contig, with hits represented as arrows that indicate the region and direction of translation. The color of the arrows corresponds to the score assigned to the hit.

Next, there are phylogenetic tree visualizations for each HMM hit. These trees are constructed by aligning the hit’s amino acid sequence with translated reference nucleotide sequences that were also identified by this HMM. By default, the reference sequences are color-coded according to their genus classification.

#### 3.2.2. Multisample Report

The pipeline offers a multisample analysis, enabling the simultaneous processing of multiple FASTQ files to provide a comprehensive overview of the entire dataset, which is useful for biodiversity analysis.

The multisample HTML report provides a summary of the alignment results to the reference database and HMM-based data scanning (Figure 3A,B). For a more detailed analysis of specific results, researchers can access the HTML report dedicated to a particular sample.

### 3.3. Applying the Pipeline to RNA Metagenomic Data

AliMarko was applied to RNA metagenomic data from bat fecal samples collected in the Moscow District, Russia, in 2015 (see Materials and Methods) [38].

Over 25 distinct viruses from animals, plants, and bacteria were identified in the dataset (Appendix A). Of these, 7 were mammalian viruses, including representatives from the *Alphacoronavirus*, *Dependoparvovirus*, *Betacoronavirus*, *Mastadenovirus*, *Sapovirus*, *Orthopicobirnavirus*, and *Alphapolyomavirus* genera. Notably, we identified contigs of insect viruses belonging to the *Iflaviridae* and *Alphatetraviridae* families, which exhibited lengths comparable to those of their respective full genomes. Notably, we found a complete genome of *Alphacoronavirus* in run SRR15525307. Furthermore, multiple fragments of *Alphacoronavirus* and *Betacoronavirus* were identified in several samples.

A *Caliciviridae* genome was detected in run SRR15534018 (Figure 4A). Using AliMarko, we identified *Caliciviridae* sequences, predicted the encoded proteins, and performed preliminary phylogenetic analysis on two coding sequences. This analysis revealed that the detected genome is from a virus that belongs to the *Sapovirus* family, but it does not cluster closely with reference genomes (Figure 4B). The genome’s length is similar to that of typical *Caliciviridae* genomes, and it also possesses a poly-A tail, which suggests it is likely a complete genome [42]. The detection of a complete *Sapovirus* genome is notable, as Sapoviruses are known to cause acute gastroenteritis in humans, highlighting the potential for transmission of these viruses between hosts [43].

Using AliMarko, we identified contigs of an *Orthopicobirnavirus*, a genus associated with gastroenteritis in humans. Two contigs were detected, one containing RNA-dependent RNA polymerase sequences (“polymerase contig”) and the other containing capsid sequences (“capsid contig”), both of which are phylogenetically close to the *Picobirnaviridae* family. Notably, phylogenetic analysis with AliMarko revealed that the contig containing the polymerase gene shows high similarity to known sequences, whereas the contig with the capsid gene exhibits low similarity.

Using a mapping module, we identified sequences in all samples that were mapped to the Moloney murine leukemia virus (*Retroviridae* family). The mapping pattern and single-nucleotide substitutions were identical across all samples, suggesting that the source of these sequences is likely a contaminant. Indeed, a previous study (Asplund et al., 2019) has indicated that Nextera kits can serve as a source of these sequences [10].

### 3.4. Comparison of HMM vs. Mapping for Viral Detection

In our study, we assessed the diversity of results from two methods. By utilizing our standard databases, we identified unique families. Notably, we processed the results to represent each family once, even if multiple contigs or genome references of a family were involved. Specifically, we focused on families present in both the MINION DB [32,33] and ICTV VMR databases. Comparing the outcomes revealed that the HMM module with MINION DB detected a greater number of families compared to the mapping module with the ICTV VMR genome collection (Figure 5). Additionally, a significant number of alignment-identified families were also detected by the HMM module. In general, the HMM module finds two times more families than the mapping module (2.4 for the dataset considered).

### 3.5. Applying the Pipeline to Diverse Set of Metagenomic Data

Our evaluation of AliMarko and Kraken2 across diverse environmental and host-associated samples—including sandflies (Phlebotomus chinensis), constructed wetland rhizospheres, and freshwater ecosystems—revealed distinct differences in viral detection capacity. By integrating results from AliMarko’s HMM and Mapping modules, the tool identified 3–18 unique viruses per sample (Appendix A), surpassing Kraken2 (PlusF database), which detected 0–5 unique viruses, with minimal overlap between the two methods (0–3 shared hits per sample). For instance, in samples such as SRR14877739 and SRR21705849, AliMarko detected 6–11 viral sequences where Kraken2 reported none, underscoring its enhanced ability to uncover viral diversity in complex metagenomes (Appendix A).

### 3.6. Validation and Performance Evaluation of AliMarko on Simulated Viral Metagenomes

To assess AliMarko’s performance, we simulated viral metagenomes using a human nasopharyngeal swab sample with manually excluded viral sequences and added sequences generated from reference genomes from eight viral families (see Methods). We generated 25 samples with varying mutation rates (SNPs: 0.01–0.17; insertions: 0.001–0.017 yielding 79.05% to 98.72% nucleotide identity to the original reference) and abundance levels (250–2250 reads yielding mean coverage depth from 0.41 to 3.68). For classification purposes, the reads and contigs from the original sample were labeled as non-viral, while the added viral sequences were labeled as viral. AliMarko’s Mapping Module (raw reads) and HMM Module (contigs) were compared to Kraken2 (core_nt database) on reads and assemblies.

Both tools demonstrated high specificity (>0.99). Kraken2’s sensitivity dropped to zero at moderate mutation rates (88% identity for reads and 93% for contigs). In contrast, AliMarko retained sensitivity even at the highest mutation levels (79% identity), detecting 6.5–8.1% of viral reads and contigs in larger samples (Figure 6). However, AliMarko’s sensitivity declined in small, highly mutated samples. Kraken2 performed better than AliMarko in detecting contigs under minimal mutation conditions.

Overall, AliMarko had a higher positive predictive value (PPV), though it decreased at extreme mutation levels, while the negative predictive value (NPV) remained stable at around 0.99 for both tools.

In summary, AliMarko is better suited for detecting highly divergent viruses, whereas Kraken2 excels in low-mutation scenarios. Both tools provide high accuracy in negative predictions.

## 4. Discussion

Viruses constitute a diverse group of biological entities, and the absence of universal single-copy genes in viruses presents a significant challenge for identifying viral sequences in metagenomic samples [1]. The low abundance of viruses in samples often results in limited concentrations of viral nucleic acids, hindering the assembly of de novo genomes for many viruses [44]. To overcome this challenge, researchers rely on more sensitive and sophisticated methodologies, including preamplification techniques and advanced bioinformatic tools, to enhance viral detection and characterization in metagenomic studies [45,46]. The sequence similarity between viruses and cellular organisms can lead to false-positive virus identifications [47]. This requires researchers to be cautious when interpreting their results and verifying their validity.

Given the complexity of identifying viral sequences, a pipeline that efficiently processes large datasets while performing essential steps to estimate the confidence of individual findings has been beneficial. AliMarko provides visualization and interpretable results, enabling experts to assess the reliability of findings and distinguish them from potential false positives.

AliMarko employs two complementary modules: reference-based mapping and de novo assembly followed by HMM-based homology analysis.

The mapping module performs aligning reads to reference sequences from the ICTV VMR database and is capable of detecting viral sequences even when the read coverage is too low to assemble contigs. Assessing the uniformity of coverage and the composition of single-nucleotide variants can help researchers evaluate mapping confidence. Uneven coverage may stem from non-specific mapping or contamination from other organisms. Additionally, identical single nucleotide variant patterns across multiple samples may indicate contamination.

During the analysis of a real dataset, we observed that Murine Leukemia Virus sequences were present in all samples, exhibiting non-uniform coverage across the genome (each sample had a distinct genomic region covered, with a consistent pattern of nucleotide substitutions). Asplund et al. [10] found that Murine Leukemia Virus sequences in their dataset were likely derived from the Nextera kit. Similarly, this suggests that the Murine Leukemia Virus sequences in our dataset may also be derived from this kit. To avoid false virus identifications, we incorporated the table of potential contaminants provided by Asplund et al. into our pipeline. As a result, users are warned about potentially contaminated findings, which are highlighted in red.

In a module complementary to the mapping, the HMM Analysis Module performs read assembly into contigs, amino acid translation, and subsequent analysis using HMM. This module is effective in identifying viral sequences assembled into contigs of several hundred bases and more. Furthermore, phylogenetic analysis enables fast interpretation of the result’s reliability and relationships with reference sequences. The structure of the phylogenetic tree can confirm the viral nature of the match and identify the phylogenetically closest reference in the database. A consistent tree structure that aligns with current taxonomy, with the contig’s leaf fitting well within a specific clade, supports a reliable conclusion. Conversely, weak phylogenetic signals, conflicting tree topology, or long branch lengths may indicate a false positive or the discovery of a novel viral representative. The HMM Analysis Module provides phylogenetic analysis of individual proteins, allowing for consideration of potential differences in mutation accumulation rates between different proteins and providing a more nuanced understanding of the evolutionary relationships between known viral strains and the virus sequence presented in the sample.

The future development of this module will focus on integrating automatic topology reliability determination and contig classification to specific clades, enhancing the pipeline’s capabilities in viral sequence analysis. It is important to note that since trees are constructed using genome fragments, their topology may deviate from the accepted taxonomy.

Among existing tools, VirID is the closest to AliMarko in terms of functionality. Both tools perform phylogenetic analysis and filtering of false positives and contaminants, indicating a shared commitment to accuracy and reliability. However, AliMarko employs a dual approach to viral sequence identification, combining contig assembly with mapping to reference genomes, which enables it to detect low-abundant viral sequences. Additionally, AliMarko has a broader scope than VirID, as it detects all types of viruses, whereas VirID is limited to *Orthornavirae* kingdom.

AliMarko is designed for Linux servers and requires only basic Linux knowledge and a Conda environment or Docker to run. The tool utilizes a set of reference genomes and HMM profiles, which can be customized as needed, allowing for flexibility in its analysis.

We applied our pipeline to metagenomic RNA sequencing data from bat fecal samples. Bat metagenomic data are characterized by a high abundance of viruses, with numerous unknown viruses identified through sequencing. Our analysis with AliMarko enabled the detection of over 25 distinct viruses, including seven mammalian viruses from different genera. Notably, we successfully identified the complete genome of a *Sapovirus* representative. This genus is known to cause acute gastroenteritis in humans.

AliMarko detected more unique viruses than Kraken2 across various environmental and host-associated samples, identifying 3–18 viruses per sample compared to Kraken2’s 0–5. AliMarko remained effective even at high mutation rates (79% identity), whereas Kraken2’s sensitivity dropped to zero at moderate mutation levels (88% identity for reads and 93% for contigs), although it performed better in low-mutation scenarios (≥98% identity). Both tools showed high specificity (>0.99), with AliMarko being better for detecting diverse viruses and Kraken2 excelling in low-mutation cases, while both provided reliable negative results.

Future developments of our pipeline will involve the incorporation of a database of non-viral sequences that are likely to be misidentified as viral (e.g., topoisomerase sequences). This will enable the detection of false mappings to viral sequences and facilitate the rooting of phylogenetic trees. Additionally, we plan to establish a web server hosting AliMarko, providing a user-friendly interface for researchers to access and utilize our pipeline.

We believe that AliMarko will enable researchers to uncover viral sequences in metagenomic datasets, including both common viruses and those that are novel or unrepresented in existing databases.

## Figures and Tables

**Figure 1 viruses-17-00355-f001:**
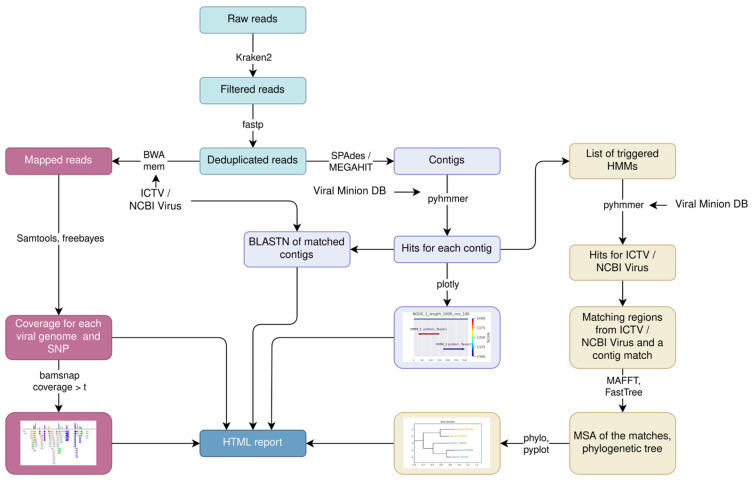
The schematic representation of the AliMarko pipeline. Two main types of analysis are performed: read mapping to reference genomes and HMM analysis of assembled contigs followed by phylogenetic analysis.

**Figure 2 viruses-17-00355-f002:**
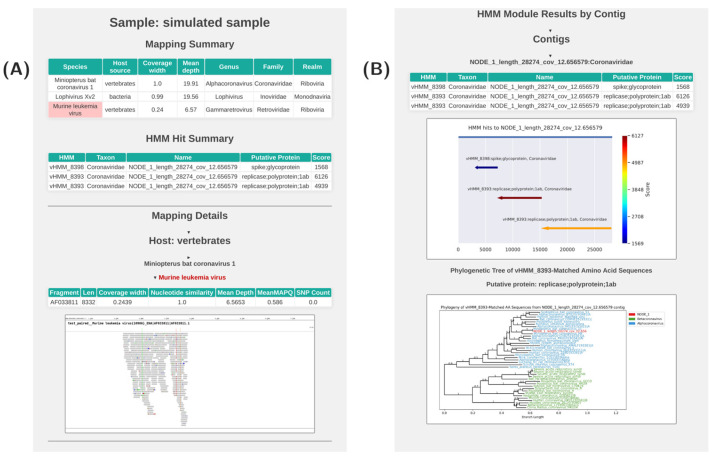
Screenshot of the AliMarko report for a sample. In the HTML report, the content of Figure B is located below the content of Figure A. (**A**)—the report contains, from bottom to top, a comprehensive table with the results of the mapping module (Mapping General Results), followed by the results of the HMM module (General HMM Results). These are followed by mapping details, which include several tabs for different assigned hosts. Upon opening the virus tab, a table displaying the mapping parameters and visualization is presented. If there is information on potential contamination for the reference genome, its name is highlighted in red. (**B**)—details for the HMM module. Each contig has its own tab, providing information on HMM hits on the contig, accompanied by a visualization of the hits. For each contig, the best blastn hit is provided following the contig data. For each hit, a phylogenetic tree is constructed based on amino acid sequences. In the tree, reference sequences are color-coded based on taxonomic classification.

**Figure 3 viruses-17-00355-f003:**
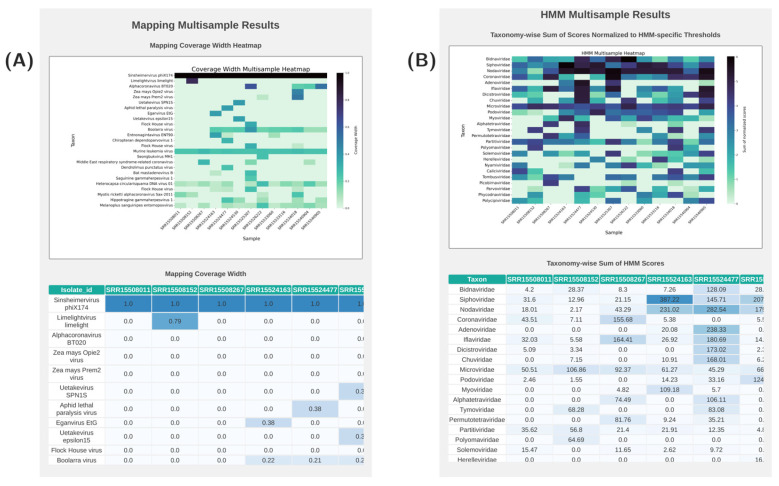
Screenshots of the multisample report of AliMarko. (**A**)—a heatmap illustrating coverages across all samples and most represented reference genomes offers a comprehensive view of the findings. Additionally, a scrollable table provides detailed coverage information, with color intensity emphasizing signal levels for clarity. (**B**)—the HMM module’s findings are detailed, featuring a heatmap displaying normalized scores and a corresponding results table.

**Figure 4 viruses-17-00355-f004:**
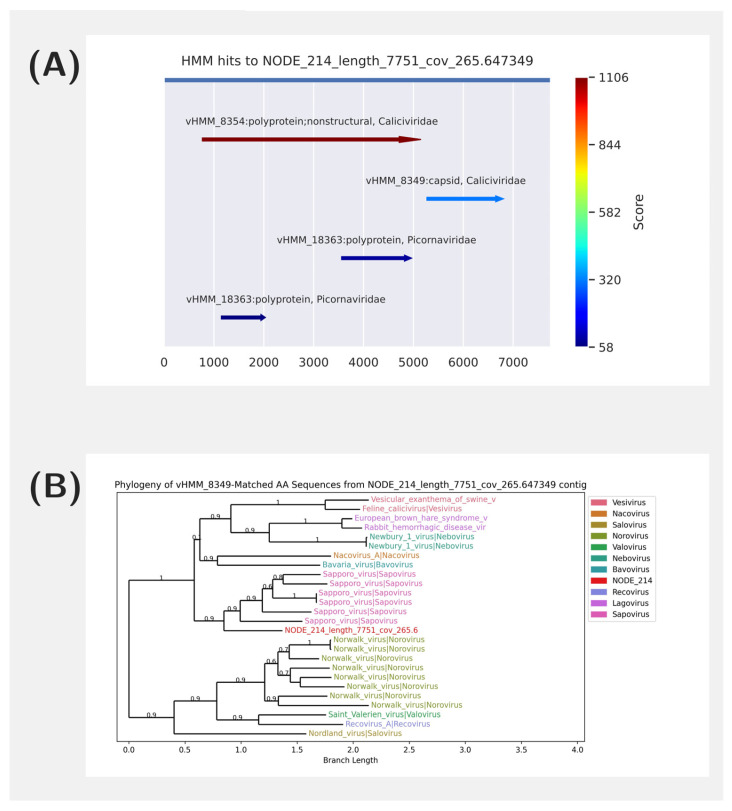
Visualizations from the AliMarko report for SRR15534018 sample. (**A**)—the visualization of HMM matches the Sapovirus contig. The matches are colored by their score. Several models matched the contig. (**B**)—the phylogenetic tree of contig of a presumably *Sapovirus* origin (see (**B**)). Sequences in the tree are colored with their taxonomic group.

**Figure 5 viruses-17-00355-f005:**
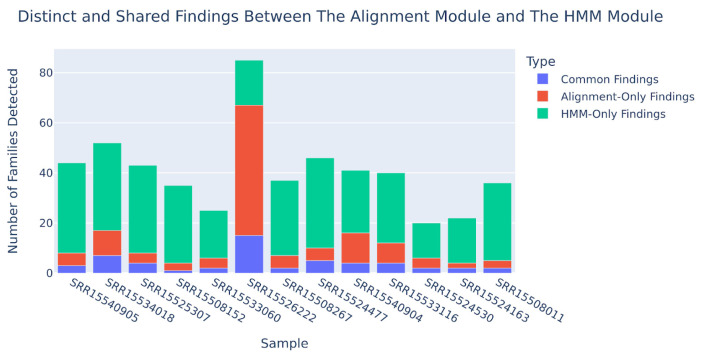
Comparison of viral families detected by the HMM module with MINION DB and the mapping module with ICTV VMR genomes collection. The barplot shows the number of unique viral families identified by each method, with the HMM module detecting a greater number of families than the mapping module.

**Figure 6 viruses-17-00355-f006:**
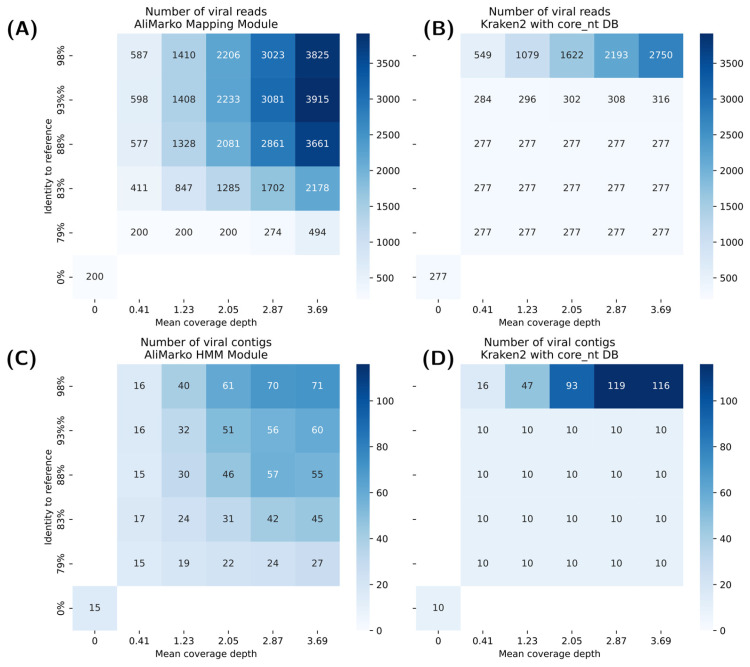
Performance comparison of AliMarko and Kraken2 across simulated viral datasets. Heatmaps illustrate the number of viral reads and contigs detected by AliMarko (**A**,**C**) and Kraken2 (**B**,**D**) under different mutation rates and abundance levels. The bottom-left cells (labeled as 0 coverage and 0% identity) represent a simulated virus-free nasopharyngeal swab sample.

## Data Availability

The study used open access data.

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
