# Peer review of "AliMarko: A Pipeline for Virus Identification Using an Expert-Guided Approach"

_viruses, 2025, doi:10.3390/v17030355_

Round 1

Reviewer 1 Report

Comments and Suggestions for Authors

In this report, the authors introduce the AliMarko pipeline to address the challenges of detecting and classifying viruses from metagenomic sequencing data. AliMarko is described as using a hybrid analysis that combines read mapping to reference databases with a hidden Markov model analyses of assembled contigs. While this represents a common mapping approach, there are some significant concerns that warrant attention:

Missing critical steps: First, the pipeline does not incorporate translated alignments, a critical step for improving sensitivity in detecting divergent viral sequences. Second, a validation step to assess the pipeline’s performance and accuracy is absent.

The authors claim that AliMarko can detect and classify both well-represented viruses and novel ones that lack representation in existing databases. However, the limited data presented in the study makes it unclear how the pipeline accomplishes this, particularly for uncovering novel or underrepresented viruses. The dataset used to demonstrate the pipeline's performance comprises only 13 samples, which is insufficient for evaluating its effectiveness. It is strongly recommended that the authors test AliMarko with more complex and diverse datasets, such as those derived from sewage, ocean water, or soil, to better assess its performance.

Read filtering step: AliMarko’s first step filters unwanted reads using Kraken with a confidence threshold of 0.7. This approach raises concerns, as such a cutoff requires a read to have a substantial amount of contaminating kmers for effective filtering, SNPs present in the read but not in the database will allow contaminating reads to pass. It is unclear if this was the intended goal or how the authors determined this parameter. Providing a justification for the choice of this cutoff would enhance the pipeline’s transparency.

Taxonomic call accuracy: The pipeline does not include additional filtration or validation steps to ensure that taxonomic assignments are not influenced by misannotated or contaminated database entries. Additionally, there is no explanation for how the pipeline resolves cases where reads or contigs map equally well to multiple database entries. Addressing this would improve confidence in the pipeline's outputs.

Validation through BLAST or equivalent: AliMarko does not incorporate a validation step using tools like BLAST, which would allow users to assess alignment accuracy and gauge the precision of taxonomic calls. Including such a step, or an equivalent, is strongly recommended to improve confidence in the pipeline’s results.

Sensitivity for low-coverage or divergent viruses: Given its current design, AliMarko may struggle to detect viruses with less than 2x coverage or those that are divergent and lack representation in existing databases. It is recommended that the authors employ or simulate datasets to show AliMarko’s performance on datasets containing virus at low coverage and divergent species.

While AliMarko offers a pipeline that integrates preexisting tools to process and classify metagenomic reads, it appears to lack innovation in its approach to identifying and classifying sequences of viral origin. Addressing the concerns highlighted above would enhance the utility, reliability, and novelty of the pipeline.

Author Response

  1. Missing critical steps: First, the pipeline does not incorporate translated alignments, a critical step for improving sensitivity in detecting divergent viral sequences. Second, a validation step to assess the pipeline’s performance and accuracy is absent.
  • 1) We agree that the use of translated amino acid sequences is an important step in virus detection because it allows the detection of more distant homology. The AliMarko pipeline uses translated sequences of assembled contigs in 6 possible reading frames for analysis using HMM models.

2) We are grateful to the reviewer for this valuable comment. In response, we have incorporated a validation step using a simulated read dataset. This addition allowed us to rigorously evaluate the accuracy, sensitivity, and specificity of our approach. The results of this validation have been included in the main text to provide a comprehensive assessment of the pipeline’s performance ( "Validation and Performance Evaluation" chapter).

  1. The authors claim that AliMarko can detect and classify both well-represented viruses and novel ones that lack representation in existing databases. However, the limited data presented in the study makes it unclear how the pipeline accomplishes this, particularly for uncovering novel or underrepresented viruses. The dataset used to demonstrate the pipeline's performance comprises only 13 samples, which is insufficient for evaluating its effectiveness. It is strongly recommended that the authors test AliMarko with more complex and diverse datasets, such as those derived from sewage, ocean water, or soil, to better assess its performance.

We appreciate the reviewer’s comment and agree that testing on more diverse datasets is important to demonstrate AliMarko’s ability to detect both well-represented and novel viruses. To address this, we’ve added an analysis of additional samples, including freshwater metagenomes, sandfly insects, and rhizosphere bulk soil metagenomes. We’ve included the results and details of this analysis in the updated main text (“Applying the pipeline to diverse set of metagenomic data”).    

  1. Read filtering step: AliMarko’s first step filters unwanted reads using Kraken with a confidence threshold of 0.7. This approach raises concerns, as such a cutoff requires a read to have a substantial amount of contaminating kmers for effective filtering, SNPs present in the read but not in the database will allow contaminating reads to pass. It is unclear if this was the intended goal or how the authors determined this parameter. Providing a justification for the choice of this cutoff would enhance the pipeline’s transparency.

We thank the reviewer for raising this important point regarding the confidence threshold used in the read filtering step.

The value of 0.7 for the confidence threshold is somewhat arbitrary. It should be considered as an acceptable starting point rather than the fine-tuned constant. Of course, the value of this (and almost any other) parameter can be adjusted in the Snakemake script. It is ultimately a researcher's responsibility to select the appropriate value for a problem under study.

To gain some understanding of the dependency of the rejection level on the confidence threshold, we performed a quick study on the simulated samples containing bacterial and human sequences. We observed that, on the data used, the threshold of 0.7 corresponds to approximately 90% of the reads we reject and 10% of the reads we use for further analysis. We used a confidence threshold of 0.7 to obtain the experimental results, presented in the paper. Given the demonstrational nature of these studies, the influence of the confidence threshold on final results was not investigated.

  1. Taxonomic call accuracy: The pipeline does not include additional filtration or validation steps to ensure that taxonomic assignments are not influenced by misannotated or contaminated database entries. Additionally, there is no explanation for how the pipeline resolves cases where reads or contigs map equally well to multiple database entries. Addressing this would improve confidence in the pipeline's outputs.

We thank the reviewer for raising these important points, which have allowed us to clarify these aspects of the pipeline.
1) To mitigate errors arising from misannotated reference sequences, we recommend using the default VMR ICTV database provided with AliMarko. This database exclusively contains viral sequences that have been curated and validated by the International Committee on Taxonomy of Viruses (ICTV), ensuring high reliability and accuracy in taxonomic assignments.

2) Regarding cases where reads or contigs map equally well to multiple database entries, we rely on the default behavior of BWA-MEM. When a read maps to multiple genomic regions with equal confidence, it is assigned a mapping quality (MAPQ) score of 0. AliMarko automatically filters out such reads during the quality filtering step prior to calculating coverage metrics. This ensures that ambiguous mappings do not influence downstream taxonomic assignments or coverage analyses.

  1. Validation through BLAST or equivalent: AliMarko does not incorporate a validation step using tools like BLAST, which would allow users to assess alignment accuracy and gauge the precision of taxonomic calls. Including such a step, or an equivalent, is strongly recommended to improve confidence in the pipeline’s results.

We appreciate the reviewer’s suggestion and have added a BLAST step to AliMarko to validate the contigs identified by HMMs. This step compares the contigs against the viral sequence database used in the analysis, helping users check alignment accuracy and the precision of taxonomic calls. In the HMM module report, the information about the best blastn hit follows the details about each contig

  1. Sensitivity for low-coverage or divergent viruses: Given its current design, AliMarko may struggle to detect viruses with less than 2x coverage or those that are divergent and lack representation in existing databases. It is recommended that the authors employ or simulate datasets to show AliMarko’s performance on datasets containing virus at low coverage and divergent species.

We appreciate the reviewer’s suggestion. In response, we have simulated a testing dataset containing viruses with low coverage and divergent sequences to evaluate AliMarko’s performance. The results of this analysis are now included in the updated text ("Validation and Performance Evaluation” chapter), providing insight into how the pipeline handles these challenging scenarios.

  1. While AliMarko offers a pipeline that integrates preexisting tools to process and classify metagenomic reads, it appears to lack innovation in its approach to identifying and classifying sequences of viral origin. Addressing the concerns highlighted above would enhance the utility, reliability, and novelty of the pipeline.

We agree with the reviewer’s observation. AliMarko is designed to combine well-validated and easily interpretable methods, prioritizing reliability and usability. While it may not introduce entirely novel algorithms, we believe this approach provides a practical and accessible tool for viral sequence analysis

Reviewer 2 Report

Comments and Suggestions for Authors

The manuscript presents the AliMarko pipeline already available on GitHub. This tool is designed for the detection of viruses from clinical specimens. For the analysis, it uses a combination of mapping reads and de novo assembly, which is useful for samples with low abundance of viruses. Furthemore, this approach helps to identify potential contaminations. The paper is clearly and comprehensively written and, in the Materials and Methods section the analysis workflow is described step by step. The results support the aim of the paper.

The manuscript rapresents a useful contribution to the field of metagenomic analysis.

Minor revision

Better explain these topics:

-       1- Is the custom database of cellular organisms closed? Is it possible to add new genomes?

-    2- Does the pipeline process fastQ files from both pair-end and single-end reads?

-        3-  Does the pipeline process fastQ files already trimmed?

-        4-  Does the pipeline identify virus recombination?

Author Response

Dear Reviewer,

We sincerely appreciate your thorough review and insightful comments on our manuscript. Please find our detailed responses in the attached document.

-       1- Is the custom database of cellular organisms closed? Is it possible to add new genomes?
Yes, the custom database of cellular organisms is not closed and can be expanded. Kraken2, which is used in the pipeline, allows users to add new genomes to the database. This flexibility enables researchers to tailor the database to their specific needs.

-    2- Does the pipeline process fastQ files from both pair-end and single-end reads?
Yes, the pipeline is designed to handle both pair-end and single-end fastQ files

-        3-  Does the pipeline process fastQ files already trimmed?
Yes, the pipeline assumes that the input fastQ files have already been trimmed. Users are expected to provide trimmed data as input to ensure optimal performance and flexibility.

-        4-  Does the pipeline identify virus recombination?
While the pipeline does not directly identify virus recombination, it provides indirect evidence that can suggest recombination events. In the HMM module, if multiple viral proteins are identified within a single contig, separate phylogenetic trees are constructed for each protein. Differences in the topology of these trees can indicate potential recombination, allowing users to make informed assumptions about recombination events.

Reviewer 3 Report

Comments and Suggestions for Authors

Strengths I noted:

The authors implemented a novel integration of two approaches (alignment to reference and de novo assembly) to virus detection, built into a pipeline with proper databases and visualizations into a potentially useful tool.

Weaknesses I noted:

  • Created a simulated dataset, but did not report detection performance compared to ground truth (major point)
  • I recommend they provide a Docker image of the tool, link on GitHub
  • Messy editing; left in instruction text, missing sentence, etc.
  • HMM hit score divided by threshold score is missing from reports in figures and Supplemental data (line 207, 221)
  • What is threshold for calling a SNP?  All illustrated examples have 0.0 in “SNP Count” column (line 213), please show an example with identified SNPs (major point)
  • Line 257: Why no overall table showing these results?  You discuss details here, let us see the table
  • Line 329: non-specific what?
  • Line 360: please perform some form of benchmarking versus existing tools

Author Response

Dear Reviewer,

We sincerely appreciate your thorough review and insightful comments on our manuscript. Please find our detailed responses in the attached document.

  • - Created a simulated dataset, but did not report detection performance compared to ground truth (major point)
    We are grateful to the reviewer for this valuable comment. In response, we have added a dedicated "Validation and Performance Evaluation" chapter to the manuscript, which includes a detailed comparison of detection performance against the ground truth.
  •  
  • - I recommend they provide a Docker image of the tool, link on GitHub
    As suggested, we have created a Docker image for AliMarko and provided the link in the GitHub repository for easier accessibility and reproducibility.
  •  
  • - Messy editing; left in instruction text, missing sentence, etc.
    Thank you for pointing this out. We have carefully reviewed the text and corrected the issues, including removing leftover instruction text and fixing missing sentences.

  •  
  • HMM hit score divided by threshold score is missing from reports in figures and Supplemental data (line 207, 221)
    We acknowledge the oversight. The Score Ratio column was initially removed from the screenshot to improve readability, and it was accidentally omitted from the sample file. This has now been corrected.
  •  
  • -What is threshold for calling a SNP? All illustrated examples have 0.0 in “SNP Count” column (line 213), please show an example with identified SNPs (major point)
    We use the default FreeBayes threshold for SNP calling. The SNP count of 0 in the examples was due to a dependency-solving bug in the version of AliMarko used to generate the example. This issue has been fixed, and we now provide examples with identified SNPs.

  • Line 257: Why no overall table showing these results? You discuss details here, let us see the table
    Thank you for the suggestion. We have added an overall table summarizing the results discussed in the text (S3 Table).
  •  
  • Line 329: non-specific what?
  • We have fixed the editing issue. The text now correctly specifies "non-specific mapping."
  • Line 360: please perform some form of benchmarking versus existing tools
    In response to this suggestion, we have added benchmarking results comparing AliMarko to Kraken2 in the updated manuscript ( "Validation and Performance Evaluation" chapter).

Round 2

Reviewer 1 Report

Comments and Suggestions for Authors

I thank the authors for addressing my suggestions. Overall, the manuscript is improved. However, I would still like to how the algorithm performs on real-world data (soil, sewer, etc.), complex metagenomic data that harbors viral signals from a wide range of viruses and different levels of coverage. I appreciate the analysis using simulated data but this represents a controlled scenario. Although valid for the purpose of testing an algorithm, it does not approximate the viral diversity of some environments where viral metagenomics would be needed. This include environmental samples as well as in rich metagenomic datasets where the presence of phages either in isolation or in prophage state are common. I strongly recommend the addition of a large and complex dataset to evaluate the performance of the algorithm. 

Author Response

Dear Reviewer,

Thank you for your valuable feedback and constructive suggestions. We greatly appreciate your comments, which have helped us improve the manuscript further.

In response to your concern regarding the evaluation of our algorithm on real-world data, we would like to highlight that, following your previous recommendation during the last revision, we analyzed datasets from three distinct environments (rhizosphere, freshwater, and insects). These results were provided in S3 Table as part of our efforts to address your earlier suggestion. However, we acknowledge that we may not have emphasized this sufficiently in the text. To improve clarity, we have now added a reference to the newly created S4 Figure, which provides a visual representation of the AliMarko results for these datasets.

Additionally, we have added information about the environmental origin of each sample to S3 Table to provide clearer context for the analyzed datasets.

Regarding phages, AliMarko is designed to detect viral signals in metagenomic data without differentiating between isolated or prophage states, enabling broad identification of viral presence across diverse samples regardless of the phages' current form.

Thank you once again for your insightful comments, which have significantly strengthened our work.

Best regards,
Nikolay Popov